# Genetic risk scores and dementia risk across different ethnic groups in UK Biobank

**Naaheed Mukadam**⬤*, **Olga Giannakopoulou, Nick Bass, Karoline Kuchenbaecker, Andrew McQuillin**

University College London Division of Psychiatry, Maple House, London, United Kingdom

* n.mukadam@ucl.ac.uk

## Abstract

### Background

Genetic Risk Scores (GRS) for predicting dementia risk have mostly been used in people of European ancestry with limited testing in other ancestry groups.

### Methods

We conducted a logistic regression with all-cause dementia as the outcome and z-standardised GRS as the exposure across diverse ethnic groups.

### Findings

There was variation in frequency of APOE alleles across ethnic groups. Per standard deviation (SD) increase in z-GRS including APOE, the odds ratio (OR) for dementia was 1.73 (95%CI 1.69–1.77). Z-GRS excluding APOE also increased dementia risk (OR 1.21 per SD increase, 95% CI 1.18–1.24) and there was no evidence that ethnicity modified this association. Prediction of secondary outcomes was less robust in those not of European ancestry when APOE was excluded from the GRS.

### Interpretation

z-GRS derived from studies in people of European ancestry can be used to quantify genetic risk in people from more diverse ancestry groups. Urgent work is needed to include people from diverse ancestries in future genetic risk studies to make this field more inclusive.

## Background

Dementia has a significant impact on those living with it, their families and society in general. It currently costs the global economy $1 trillion annually and these costs are likely to increase as life expectancy increases [1]. There is therefore an urgent need to understand mechanisms of disease development and to develop preventative strategies. The most common genetic risk factor for dementia is Apolipoprotein E (APOE), with the ε2 allele being protective against

**Data Availability Statement:** The data that support the findings of this study (UKB project ref 40055) contain sensitive and potentially identifying information and therefore not suitable for sharing.

This is due to the terms of the UK Biobank data use. Further details on the study are available from the corresponding author, (NM) upon reasonable request. For access to the data please contact access@ukbiobank.ac.uk.

**Funding:** NM is funded by an Alzheimer's Society fellowship (AS-SF-18b-001). The funders had no role in study design, data collection and analysis, decision to publish, or preparation of the manuscript.

**Competing interests:** The authors have declared that no competing interests exist.

Alzheimer's dementia and the ε4 allele greatly increasing risk [2]. The APOE gene is also associated with the risk of vascular [3], Lewy Body [4] and other forms of dementia [5]. Genetic risk scores (GRS) are a way of summing the genetic risk that an individual carries for a particular disease. They are developed following Genome-Wide Association Studies (GWAS) that test the association between genetic variants–usually Single Nucleotide Polymorphisms (SNP) and phenotypes. They have become increasingly important in quantifying risk for dementia [6, 7] with good ability to predict both dementia and mild cognitive impairment [8] in people of European origin. In general, scores which use data from the whole genome are referred to as polygenic risk scores and those which use only SNP with genome wide significance are referred to as Genetic Risk Scores (GRS) [9]. Being able to quantify the genetic risk for a disease is important in being able to predict who is at greater risk and to potentially intervene earlier to prevent the disorder. However, around 80% of GWAS are restricted to people of European origin [10, 11] despite the fact that only 11.5% of the world population is of European origin [12]. Using GRS from GWAS conducted in European origin samples to predict disease status in diverse ethnic groups in general could be less accurate than predicting phenotypes in those of European origin [13] due to differences in allele frequencies and patterns of linkage disequilibrium, potentially leading to widening health inequalities. Although conducting GWAS in diverse groups could lead to identification of new disease variants and would be more equitable, this requires tens of thousands of cases and controls and is resource intensive [14].

Previous large GWAS in European origin participants have identified a number of SNPs associated with dementia, with the SNPs in turn being part of genes identified as important in brain amyloid clearance or the immune system [15]. GRS constructed using these risk alleles, including APOE have been used to predict dementia and are correlated with cognitive test scores, cortical thickness [16] and hippocampal [17, 18] and amygdala [17] volume as well as atrophy in basal forebrain and temporal atrophy [19]. So far, this quantification has been limited to people of European origin with some promise in validating these genetic risk scores in a relatively small sample of East Asians [20]. Another study found less robust prediction of dementia propensity scores in non-Hispanic Black participants in the USA compared with non-Hispanic White participants using a genetic risk score incorporating APOE and nine other SNPs [21]. Established genetic risk factors for dementia from GWAS in people of European ancestry were associated with learning and delayed recall in a sample of around 900 South Asians [22] but have not been tested against a clinical diagnosis of dementia. Assumptions that GRS developed in European origin participants will not be valid for predicting dementia in people from diverse ancestries has meant that people from diverse ancestries have been excluded from investigations of interactions between genetic risk and environmental risk factors [23] and cognitive trajectories in later life [24], all with the potential to improve dementia prevention and treatments.

In this study we aimed to establish how well GRS for dementia developed in European ancestry participants could predict dementia and other associated factors in the two main minority ethnic groups in the UK. We reasoned that if ethnicity did not modify the association between GRS and dementia risk then this could lead to greater inclusion of people from diverse ethnic groups in future epidemiological and interventional studies.

Our aims were to:

1. Establish frequency of APOE variants across ethnic groups

2. Assess the effect of APOE on risk of dementia and test for an interaction between ethnic group and APOE on dementia risk

3. Assess how well Genetic Risk Score (GRS) based on data from people of European origin predicts dementia risk in a diverse group and test whether ethnic group modifies this association

4. Assess association of GRS with other traits linked to dementia, including cognition

## Methods

### Sample

The UK Biobank is a population-based cohort of more than 500,000 UK-based participants, with differing socioeconomic and ethnic characteristics [25], all aged 40 and over at registration from 2006 to 2010. Participants completed demographic and health questionnaires, interviews and blood, urine and saliva tests. UK Biobank also links participants to their existing health records, such as those from general practice (GP), hospitals and those collected centrally (cancer and death statistics for instance). The UK Biobank team assert that all procedures contributing to this work comply with the ethical standards of the relevant national and institutional committees on human experimentation and with the Helsinki Declaration of 1975, as revised in 2008. The UK Biobank received approval from the National Information Governance Board for Health and Social Care and the National Health Service North West Multicentre Research Ethics Committee. All participants provided informed consent through electronic signature at baseline assessment. We obtained internal ethical approval from UK Biobank for our proposal (ref 40055).

### Genotyping and quality control procedures

Genome-wide genotyping was performed on all UK Biobank participants using the UK BiLEVE or UK Biobank Axiom Array. Approximately 850,000 variants were directly measured, with > 90million variants imputed using the Haplotype Reference Consortium and UK10K + 1000 Genomes reference panels [26]. UK Biobank conducted initial quality control procedures and flagged samples that were outliers in terms of heterozygosity or missing rates. They also provided genetic principal components values generated from all valid samples and a list of samples with sex discrepancies. We conducted quality control procedures for European and non-European samples separately, including estimates of relatedness between individuals. We excluded those with sex discrepancies and people who had a kinship coefficient of 0.0625 or greater (equivalent to a third degree relative or closer) [27] with another participant. We excluded SNPs with a Minor Allele Frequency (MAF) of <1% and Info score of <0.9.

### APOE

We used two SNPs rs429358 and rs7412 as markers for APOE [28]. These SNPs were directly genotyped in UK Biobank. We used combinations of the polymorphisms at these two SNPs to determine APOE alleles [29] as described in S1 Table in S1 File.

### Genetic risk score

We extracted genotypes for SNPs that have previously shown an association with dementia risk in a large Genome Wide Association Study in European ancestry individuals [15]. We specifically only included SNPs that showed an association with genome wide significance ($p<5\times10^{-8}$) across both the training and validation datasets, in a previous study which did not include UK Biobank [15]. For each individual in the UK Biobank, we calculated a GRS by

multiplying the genotype dosage of each risk allele for each variant (except APOE SNPs) by its respective weight (log odds ratio (OR) for dementia), and then summing across all variants and dividing by number of variants [30] (see supplementary text for formula) using the—score function in Plink 2.0 [31]. APOE genotype was calculated as detailed above and number of ε2/ε4 was multiplied by the log OR for dementia associated with ε2/ε4 respectively. This was then added to the GRS calculated for other SNPs. The weighting for each SNP including APOE was as reported by Lambert and colleagues [15]. Details of SNPs and log odds ratios used is in S2 Table in S1 File. The genetic risk scores were then z-standardized across the whole sample and this was used as our main exposure variable. We repeated all analyses using a z-standardised GRS calculated from all SNPs but excluding APOE.

## Main outcome

We determined the diagnosis of dementia from self-report or linked electronic health data in the UK Biobank [32]. The positive predictive values (PPVs) for the diagnosis of all-cause dementia were 86.8% for primary care, 87.3% for hospital admissions and 80.0% for mortality data respectively and 82.5% across all datasets when compared against blinded clinician diagnosis from notes using standardised diagnostic criteria [33]. This indicates that all-cause dementia as identified within UK Biobank is a valid diagnosis. We chose all-cause rather than Alzheimer's disease specifically because the number of dementia cases specified as Alzheimer's was relatively low and most dementia types share common genetic risk, particularly APOE, as outlined above. ICD-10 codes used to identify dementia cases are as described previously [32] and included F00, F01, F02, F03 and G30 subcategories.

## Secondary outcomes

Family history of dementia: Participants self-reported whether their father or mother had been diagnosed with dementia. We combined information from these questions to designate whether participants had a positive family history of dementia, scoring zero for no family history, one for one parent and two if both parents had dementia.

Reaction time: Participants were shown two cards at a time on the touchscreen and instructed to press the button on the button box as quickly as possible when the symbols on the cards match. The exercise involved 12 pairs of cards. Mean time taken to correctly identify matches was calculated by UK Biobank, using data from all rounds in which both cards matched.

Brain volume: Over 40,000 UK Biobank participants have had three structural brain MRI scans; T1, T2 fluid attenuation inversion recovery (FLAIR) and susceptibility-weighted MRI (swMRI). T1 scans allow precise volumetric measures of the whole brain, as well as specific cortical and subcortical regions. An automated processing pipeline for brain image analysis and quality control was established for UKB at the University of Oxford's Wellcome Centre for Integrative Neuroimaging (WIN/FMRIB) [34]. This pipeline is primarily based around FSL (FMRIB's Software Library), and other packages such as FreeSurfer [35]. We used hippocampal and amygdala volume in mm$^3$ (combining volumes for left and right structures) as calculated from T1 images carried out at visit 2(2014), using Automatic Segmentation (ASEG) [36], as two of our secondary outcome measures, adjusting for total brain volume which was normalised for head size as well as age and sex.

## Ethnicity and genetic ancestry

All participants self-identified their ethnicity using the UK Census categories. For this study we combined ethnicities into White (British and Irish); South Asian (Asian Bangladeshi,

Indian, Pakistani); or Black or Black British (Black African, Black Caribbean, Black British, Black Other), based on research showing clustering of low frequency genetic variants in people originating in the same geographic location [37]. We chose self-defined ethnicity for our primary analysis as this is more easily available for most studies and is a practical choice for grouping. We focused on these three ethnic groups as they were the largest in the sample and in the UK overall. Only those with a valid ethnicity in their records were included in our analyses as this was our main exposure variable of interest. We repeated all analyses using genetically defined European, South Asian and African ancestry. We conducted principal component (PC) analysis in order to classify participants into European, African, South Asian and admixed-ancestry groups. First, we estimated PCs for a subset of non-European ancestry participants using PC-AiR implemented in the GENESIS package, that has been optimized for admixed-ancestry datasets [38]. We assigned the samples into ancestry groups, based on visual inspection of the PCs as described previously [39]. Further details of this PC analysis are also shown in the supplementary material.

## Covariates

We considered the following variables based on data which participants gave at baseline, as the association between genotype and dementia might be confounded by these.

Age–We adjusted for baseline age in all models.

Sex–participants self-identified as male or female and we adjusted for sex in all models.

The first 10 genetic principal components, as calculated separately by UK Biobank for European samples and by our group, for non-European ancestry participants, was included in all models testing the association between genetic ancestry and dementia to adjust for genetic variability.

## Statistical analysis

We used Stata 16.0 for all statistical analyses. First, we summarised age, sex, APOE variants, other genetic variants of interest, brain volume measures and mean GRS across ethnic groups.

To identify main effects of exposures, for self-defined ethnicity regression analyses, we first ran regression models with ε2 and ε4 alleles as exposures with age, sex and ethnicity as covariates. We then repeated the analyses, including an interaction term for ethnicity and ε2/ε4 alleles respectively. To assess how well GRS predicted dementia across self-defined ethnic groups, we conducted a logistic regression with all cause dementia as the outcome and z-standardised GRS as the exposure, including age, sex and ethnic group as covariates. We then tested how well z-GRS predicted family history of dementia using logistic regression, adjusting for age, sex and ethnic group. To test for the association between z-GRS and cognition we used linear regression with reaction time as the outcome variable and z-GRS as exposure, adjusting for age, sex and ethnicity. We also tested the association between z-GRS score and amygdala and hippocampal volumes, using linear regression, adjusting for age, sex, ethnicity and total brain volume. We repeated these analyses, including an interaction term between z-GRS and ethnicity. We then conducted all these analyses using z-GRS excluding APOE as the exposure, conducting analyses to identify main effects then repeating analyses and including interaction terms between z-GRS and ethnicity. All analyses were then repeated using genetically defined ancestry groups and including the first 10 principal components derived for each ancestry group separately as well as age, sex and genetic ancestry groups as covariates in all models. We tested interaction effects between self-defined ethnicity/genetically determined ancestry and APOE allele status and z-GRS in turn.

**Table 1. Characteristics of included participants by self-defined ethnicity.**

| Characteristic | | White British (N = 364,879) | South Asian (N = 6,665) | Black (N = 6,927) |
|---|---|---|---|---|
| Age (mean, SD) | | 56.8 (8.0) | 53.4 (8.5) | 51.9 (8.1) |
| N(%) female | | 196,247 (53.8) | 3,054 (45.8) | 3,904 (56.4) |
| Family history of dementia N(%) | None | 318,191 (87.2) | 6,357 (95.4) | 6,519 (94.1) |
| | One parent | 44,789 (12.3) | 292 (4.4) | 394 (5.7) |
| | Both parents | 1,899 (0.5) | 16 (0.2) | 14 (0.2) |
| ε2 allele N(%) | 0 | 317,779 (87.1) | 6,154 (92.3) | 5,852 (84.5) |
| | 1 | 44,802 (12.3) | 501 (7.5) | 992 (14.3) |
| | 2 | 2,298 (0.6) | 10 (0.2) | 83 (1.2) |
| ε4 allele N(%) | 0 | 260,152 (71.3) | 5,440 (81.6) | 4,101 (59.2) |
| | 1 | 96,006 (26.3) | 1,160 (17.4) | 2,457 (35.5) |
| | 2 | 8,721 (2.4) | 65 (1.0) | 369 (5.3) |
| Reaction time in seconds—mean(SD) | | 556 (114) | 614 (157) | 632 (177) |
| Dementia cases N(%) | | 4,729 (1.3) | 73 (1.1) | 87 (1.3) |
| Mean z-standardised GRS score including APOE (SD) | | 0.00(1.00 | -0.15 (0.79) | 0.23 (1.19) |
| Mean z-standardised GRS score excluding APOE (SD) | | 0.01 (1.00) | 0.11 (0.97) | -0.50 (0.83) |
| **Imaging sub-sample** | | **N = 32,626** | **N = 328** | **N = 243** |
| Hippocampal volume—left plus right(Mean(SD) in mm$^3$) | | 8066.6 (805.0) | 7953.4 (849.2) | 7956.1 (746.1) |
| Amygdala volume—left plus right(Mean(SD) in mm$^3$) | | 3289.7 (426.0) | 3292.0 (468.2) | 3252.9 (398.1) |
| Brain volume, normalised for head size (Mean(SD) in mm$^{3*}$10$^6$) | | 1.5(0.1) | 1.5 (0.1) | 1.5 (0.1) |

## Results

There were 502,516 participants in the whole sample, of whom 487,395 had valid genotyping data. After excluding outliers for heterozygosity (n = 115) and individuals who were related to others in the sample (81,889 White, 699 South Asian and 318 Black participants), and 25,903 people of other ethnicities, we included 364,879 self-defined White British, 6665 South Asian and 6927 Black participants. Seventeen of the twenty-one SNPs of interest were directly genotyped in UK Biobank and the remaining were imputed. All SNPs had an Info score of over 0.9 and a Minor Allele Frequency (MAF) of >1%, with similar values for each in those of European and non-European ancestry (S3 Table in S1 File). Overall, 4,729 (1.3%) White British, 73 (1.1%) South Asian and 87 (1.3%) of Black participants were diagnosed with dementia.

Baseline characteristics are shown in Table 1. Black and South Asian participants were on average younger than the White participants and the proportion of women was highest in the Black group compared to the White and South Asian groups. Self-reported family history of dementia was much lower in the South Asian and Black participants compared to the White participants.

### APOE frequencies

South Asian participants had the lowest frequency of ε2 alleles (7.5% heterozygous and 0.2% homozygous carriers) and Black participants had the highest frequency (14.3% heterozygous and 1.2% homozygous carriers) compared to the White participants (12.3% heterozygous and 0.6% homozygous carriers). The ε4 allele was also most common in Black participants (35.5% heterozygous; 5.3% homozygous carriers vs 17.4% and 1.0% in South Asians and 26.3% and 2.4% in White participants). Mean z-standardised GRS (z-GRS) score including APOE was lower in South Asians and higher in Black participants and the score excluding APOE was lower in South Asian and Black participants compared to White participants.

**Table 2. Dementia risk associated with genotype by self-defined ethnicity.**

| Exposure | | Main effect | | | | Interaction effect, South Asian participants, OR (95%CI, p-value) | Interaction effect, Black participants, OR (95%CI, p-value) |
|---|---|---|---|---|---|---|---|
| | | OR* | LCI | UCI | p value | | |
| ε2 alleles | 0 | Reference | | | | OR 1.46 (0.58–3.66, 0.336) | OR 0.95 (0.47–1.95, 0.896) |
| | 1 | 0.54 | 0.48 | 0.60 | <0.0001 | | |
| | 2 | 0.56 | 0.35 | 0.89 | 0.014 | | |
| ε4 alleles | 0 | Reference | | | | OR 0.92 (0.60–1.39, 0.676) | OR 0.99 (0.72–1.37, 0.974) |
| | 1 | 2.59 | 2.44 | 2.75 | <0.0001 | | |
| | 2 | 8.47 | 7.68 | 9.34 | <0.0001 | | |
| z-GRS with APOE—effect per SD increase | | 1.73 | 1.69 | 1.77 | <0.0001 | OR 0.97 (0.76–1.20, 0.687) | OR 0.99 (0.83–1.17, 0.864) |
| z-GRS without APOE–effect per SD increase | | 1.21 | 1.18 | 1.24 | <0.0001 | OR 0.82 (0.65–1.05, 0.113) | OR 0.94 (0.72–1.23, 0.620) |

*Odds ratio of dementia, adjusted for age, sex and ethnic group. SD = Standard deviation LCI = Lower confidence interval UCI = Upper Confidence interval.
Interaction tested between ethnicity and APOE allele or z-GRS association with dementia.

## Regression analyses

Results from the regression analyses for genetic risk factors are shown in Table 2. Logistic regression showed that the ε2 allele roughly halved dementia risk but not in a dose dependent way (OR for one ε2 allele 0.54 and for two 0.56). There was no evidence to support an interaction effect between ethnicity and ε2 allele in terms of dementia risk (p-values for interaction >0.05 for both South Asian and Black participants). One ε4 allele more than doubled dementia risk (OR 2.59, 95% CI 2.44–2.75) and two ε4 alleles increased risk by more than eight-fold (OR 8.47, 95% CI 7.68–9.34). There was no evidence to support an interaction between ethnicity and ε4 (p value for interaction >0.05 in both minority ethnic groups).

Per standard deviation (SD) increase in z-GRS including APOE, the odds ratio for dementia was 1.73 (95%CI 1.69–1.77) with no evidence for an interaction effect with ethnic group. Z-GRS which excluded APOE also increased risk of dementia (OR per SD increase 1.21, 95% CI 1.18–1.24) with no evidence for an interaction effect.

Table 3 shows results of secondary outcome regression analyses. Z-GRS including APOE was associated with family history of dementia (OR 1.32 per SD increase, 95% CI 1.31–1.33) with no evidence for interaction with South Asian ethnicity (p = 0.086) and a slightly smaller effect in people who self-defined as Black (p = 0.013). Each increase of one standard deviation

**Table 3. Regression analyses for secondary outcomes with z-GRS including APOE as exposure.**

| Dependent variable | Main effect | | | | Interaction effect, South Asian participants, Coeff/OR (95%CI, p-value) | Interaction effect, Black participants, Coeff/OR (95%CI, p-value) |
|---|---|---|---|---|---|---|
| | Coeff/OR* | LCI | UCI | p-value | | |
| Family history of dementia | 1.32 | 1.31 | 1.33 | <0.0001 | 0.89 (0.78–1.02, 0.086) | 0.90 (0.83–0.98, 0.013) |
| Reaction time | 0.14 | -0.21 | 0.49 | 0.44 | 1.58 (-1.92 to 5.08, 0.377) | 1.24 (-1.06 to 3.52, 0.291) |
| Hippocampal volume | -7.20 | -14.89 | 0.48 | 0.07 | -8.69 (-101.54 to 84.15, 0.854) | 10.92 (-64.04 to 85.85, 0.775) |
| Amygdala volume | -2.39 | -6.33 | 1.54 | 0.23 | -28.56 (-76.14 to19.03, 0.240) | -1.58 (-39.99 to 36.83, 0.936) |

*Coefficient for linear regression and Odds ratio for logistic regression after age, sex and ethnic group adjustment (and brain volume for hippocampal and amygdala volume regressions), LCI = Lower confidence interval, UCI = Upper confidence interval. Interaction tested between self-defined ethnic group and exposure and dementia risk

of z-GRS increased reaction time by 0.14 seconds but this was not statistically significant (p = 0.44). There was a trend for each standard deviation increase of z-GRS to decrease hippocampal and amygdala volume, but confidence intervals were wide and there was no evidence of an interaction with ethnic group for any of the analyses. Associations with secondary outcomes were much smaller using the z-GRS which excluded APOE, but with no evidence of an interaction effect in minority ethnic groups (S4 Table in S1 File).

When participants were grouped by genetic ancestry, we included 368,277 White, 7869 South Asian and 8413 Black participants. Baseline characteristics and results of regression analyses when participants were grouped by genetic rather than self-defined ethnicity, were similar to main analyses, as shown in the supplementary material (S5-S7 Tables in S1 File).

## Discussion

To our knowledge this is the first investigation of the association between genetic risk score developed in people of European ancestry and dementia diagnosis across different ethnic groups in the UK, including minority ethnic groups. We find that APOE alleles distribution varies by ethnic groups, but the alleles have a similar association with dementia across all ethnic groups. This contrasts with previous findings that the APOE ε4 allele is not associated with dementia in African Americans [40] and older Nigerians [41, 42] although others have found a significant association in African Americans [43]. We replicate findings that APOE impact on dementia risk is not affected by global ancestry [44]. Our estimates of odds ratios associated with the APOE ε4 alleles is very similar to previous findings, indicating validity [15]. Distribution of ε2 and ε4 alleles is similar to other surveys, although there is great variability within ethnic groups [45]. However, our findings, particularly of higher ε4 frequency in those of Black ethnicity suggests this genetic risk should be more widely considered in these populations as this could have public health implications. GRS including APOE predicted dementia in all ethnic groups and was also associated with family history of dementia. There was also some indication of association with secondary outcomes, but these analyses had less power. GRS excluding APOE was also associated with dementia but with smaller coefficients and no evidence for modification of effect by ethnic group. Accuracy of genetic risk and its association with dementia is improved in all groups by including APOE.

Those from minority ethnic groups were less likely to report family history of dementia. This may be due to under-diagnosis of dementia in their parents, due to bias within the healthcare system or reluctance of people from minority ethnic people to seek a dementia diagnosis or to report it if given a diagnosis [46]. A greater proportion of Black participants had the ε4 allele compared to the White and South Asian participants, so even though the association of this allele with dementia risk was similar to the White population, Black people had a higher genetic burden, which could explain previous findings of an increased risk of dementia in Black participants in the UK Biobank [47].

Mean z-GRS varied by ethnic group. However, differences in average GRS caused by population differences in allele frequencies are expected [48] and do not necessarily mean there are corresponding group differences in risk. Our results align with a previous study showing that allele frequencies of key SNPs associated with dementia are different in South Asians compared to the European origin populations from which GRS are derived [22]. The same study also found that a relatively small percentage of the variance of cognitive scores in South Asian participants was explained when using GRS that did not include APOE, which also supports our findings for secondary outcome measures.

The strengths of this study are the large sample, including a relatively large number of people of self-defined South Asian and Black ethnicity. However, we acknowledge that the ratio of

White to minority ethnic participants was high. Follow up time was almost 15 years and the total number of dementia cases was high, allowing for robust estimates. However, there was a relatively small number of dementia cases in minority ethnic groups and it is possible interaction analyses were therefore underpowered. In addition, not all participants in the sample had linked hospital and primary care records so some diagnoses of dementia may have been missed. The number of people with brain scans was relatively low which means we had reduced power to detect an association between z-GRS and hippocampal and amygdala volumes. We also acknowledge that self-defined ethnicity may be associated with other societal factors that could influence associations with dementia.

Historically, people of non-European ancestry have been excluded from genetic studies as including diverse groups is more methodologically and computationally demanding and it was thought that the difference in patterns of linkage disequilibrium and allele frequencies may lead to population stratification and bias [49]. This has meant that most advances in understanding the genetics of dementia has come from those of European origin and people from other ethnic groups have been excluded. A lack of disease modifying treatments and growing evidence for dementia prevention [1] have meant there is increasing interest in recruiting those with higher genetic risk of dementia for investigation of disease mechanisms and targets for prevention and treatment. People from minority ethnic groups tend to be under-represented in dementia trials [50]. Not only does this mean we have limited understanding of whether treatments are effective in everyone, but it also misses the opportunity to study diverse mechanisms of disease. To advance our understanding of this complex disorder, it is imperative that we quantify genetic risk and link this with phenotypes in a diverse range of people.

Overall, our study shows that GRS for dementia generated from GWAS in European origin samples is associated with dementia diagnosis in those from diverse ancestries even when the GRS did not include APOE. This indicates that all variants with strong evidence of association with dementia discovered in those of European ancestry can be used to quantify genetic risk in those from more diverse ancestries and exclusion of people from diverse ancestries from future dementia risk studies is not justifiable. Previous work has shown that disease prediction is improved when multi-ethnic samples are used to develop GRS [51]. Studying genetic risk of diseases in diverse-ancestry groups also has the potential to identify new genetic risk loci and further our understanding of disease [52]. It is therefore imperative that future GWAS include people from diverse backgrounds in order to make this field more inclusive.

## Supporting information

**S1 Fig.**
(TIF)

**S2 Fig.**
(TIF)

**S3 Fig.**
(TIF)

**S1 File.**
(DOCX)

**S2 File.**
(PDF)

## Author Contributions

**Conceptualization:** Naaheed Mukadam, Olga Giannakopoulou, Nick Bass, Karoline Kuchenbaecker, Andrew McQuillin.

**Data curation:** Naaheed Mukadam, Olga Giannakopoulou.

**Formal analysis:** Naaheed Mukadam.

**Funding acquisition:** Naaheed Mukadam.

**Investigation:** Naaheed Mukadam.

**Methodology:** Naaheed Mukadam, Olga Giannakopoulou, Nick Bass, Karoline Kuchenbaecker.

**Project administration:** Naaheed Mukadam, Andrew McQuillin.

**Resources:** Naaheed Mukadam.

**Supervision:** Nick Bass, Karoline Kuchenbaecker, Andrew McQuillin.

**Writing – original draft:** Naaheed Mukadam.

**Writing – review & editing:** Naaheed Mukadam, Olga Giannakopoulou, Nick Bass, Karoline Kuchenbaecker, Andrew McQuillin.

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
