## [Decision Letter · Decision Letter 0]

23 Aug 2022

PONE-D-22-19235Genetic risk scores and dementia risk across different ethnic groups in UK BiobankPLOS ONE Dear Dr. Naaheed Mukadam,

Thank you for submitting your manuscript to PLOS ONE. After careful consideration, we feel that it has merit but does not fully meet PLOS ONE’s publication criteria as it currently stands. Therefore, we invite you to submit a revised version of the manuscript that addresses the points raised during the review process.

ACADEMIC EDITOR:Please address the concerns raised by Reviewer #1, especially by providing the AUROC results either in main text or in supplementary material.

Please submit your revised manuscript by Oct 7, 2022. If you will need more time than this to complete your revisions, please reply to this message or contact the journal office at plosone@plos.org. Please include the following items when submitting your revised manuscript:A rebuttal letter that responds to each point raised by the academic editor and reviewer(s). You should upload this letter as a separate file labeled 'Response to Reviewers'.A marked-up copy of your manuscript that highlights changes made to the original version. You should upload this as a separate file labeled 'Revised Manuscript with Track Changes'.An unmarked version of your revised paper without tracked changes. You should upload this as a separate file labeled 'Manuscript'.

We look forward to receiving your revised manuscript.

Kind regards,

Jeong-Sun Seo

Academic Editor

PLOS ONE

Journal Requirements:

"NM is funded by an Alzheimer’s Society grant (AS-SF-18b-001). NM, NB, KK and AM are supported by the NIHR BRC".

 "NM is funded by an Alzheimer’s Society fellowship (AS-SF-18b-001). The funders had no role in study design, data collection and analysis, decision to publish, or preparation of the manuscript".

Reviewers' comments:

Reviewer's Responses to Questions

**Comments to the Author**

1. Is the manuscript technically sound, and do the data support the conclusions?

Reviewer #1: No

Reviewer #2: Yes

2. Has the statistical analysis been performed appropriately and rigorously? 

Reviewer #1: No

Reviewer #2: Yes

3. Have the authors made all data underlying the findings in their manuscript fully available?

Reviewer #1: Yes

Reviewer #2: Yes

4. Is the manuscript presented in an intelligible fashion and written in standard English?

Reviewer #1: Yes

Reviewer #2: Yes

5. Review Comments to the Author

Reviewer #1: In the present manuscript, the authors used microarray data comprising 850,000 variants and more than 90 million variants were used for downstream analysis after imputation. After QC, they extracted previously identified dementia risk associated SNPs from previous GWAS (Lambert et al., 2013, Nature Genetics) which were not included in the UK Biobank set, and calculated GRS using those SNPs and weight in addition to APOE alleles. Since OR of APOE for dementia is extremely high, they used logOR for APOE alleles. Then they prepared 1) GRS with APOE alleles followed by z-standardization across whole sample (z-GRS with APOE), and 2) GRS without APOE alleles followed by z-standardization (z-GRS without APOE). They performed age, sex and ethnic group adjusted logistic regression analysis to investigate the effect of GRS and APOE alleles for dementia risk. Authors showed that significant increase dementia risk for APOE ε4 alleles, and the effects per SD increase of z-GRS with APOE and without APOE were significantly associated with increasing dementia risk, without significant ethnic group interaction effect. Authors also demonstrated association between SD increase of z-GRS and other features of dementia including family history, reaction time, and hippocampal and amygdala volumes, and showed significant association between z-GRS and family history. Also in this case, they have shown the effects of z-GRS for secondary outcomes had little or no significant interactions with other ethnic groups.

Overall, the manuscript falls below the level of publication in PLOS one, and authors are required to improve this study by referring the designs of many other research studies with strong understanding of genetics.

Major points

1)Main purpose of this study was to assess relevance of PRS model constructed from GWAS using Europeans. The way of showing applicability of European GWAS derived GRS in other ethnic groups is difficult to be agreed. Authors have measured interaction effect of ethnicity in logistic regression using z-GRS for dementia risk. This analysis presents whether ethnic differences interacted with z-GRS for dementia risk, not telling transferability of European GRS to South Asian and African groups. So this cannot support authors’ research questions. Instead, applying GRS model to South Asian and African separately, and then measure proportion of dementia cases or odds ratio by quantile groups, or area under receiver operating characteristics (AUROC) which represent classification power of the model.

2)Also authors conducted z-standardization to have z-score of GRS across whole sample in the UK Biobank set. This approach may diminish ethnic differences due to skewed proportion of White British (approximately 96% of total). It seems not appropriate to use z-standardization across whole sample for assessing transferability of European GRS to other ethnic groups.

3)To precisely assess applicability of European PRS models, various PRS calculation methods can be used. Authors can clump using linkage disequilibrium then adjusted p-value thresholds of European GWAS to construct various models (Pruning and thresholding) or leveraging genome-wide SNPs and adjust and shrink effect sizes using Bayesian models (PRScs or EB-PRS).

4)Authors are strongly required to graphically illustrate their result regarding their results.

Minor points

1)Authors need to change the terms for APOE alleles; 0, 1 and 2 into non-carrier, heterozygous ε2 allele carrier, and homozygous ε2 allele carrier, respectively.

2)In the Table 2, APOE allele non-carriers (0 in this paper) were labelled as “Reference” but this is not necessarily presented in the table, and exact p-values are missing for interaction effects.

3)Authors have used threshold for Info score of 0.9, and this might be too high. Authors may present references for justifying this threshold.

4)Authors are required to proofread the manuscript to avoid awkward descriptions.

5)Result section should be divided into subtitles for better readability.

 

Leonenko et al., 2021, Nature communications

https://www.nature.com/articles/s41467-021-24082-z

Najar et al., 2021, Alzheimers Dment (Amst)

https://www.ncbi.nlm.nih.gov/pmc/articles/PMC7821873/

Baker and Escott-Price, 2020, Front. Digit. Health

https://www.frontiersin.org/articles/10.3389/fdgth.2020.00014/full

Fritsche et al., 2021, PLOS Genetics

https://journals.plos.org/plosgenetics/article?id=10.1371/journal.pgen.1009670

Ge et al., 2022, Genome Medicine

https://genomemedicine.biomedcentral.com/articles/10.1186/s13073-022-01074-2#Sec14

Reviewer #2: Through Genome-Wide Association Studies (GWAS), researchers have developed Genetic Risk Scores (GRS) for numerous diseases. However, these studies are mostly biased towards European white races, so their effectiveness in other populations is not certain. Recognizing this problem, the authors verified the GRS developed for dementia in other races other than European white.

Such verification is expected to improve reliability in the use of the developed GRS as well as to enable the classification of GRS that cannot be used for various races. Therefore, I agree to accept this research for publication on Plos One.

I found a few minor things that needed to be corrected as described below.

1. The tables in the paper are analyzed for self-defined ethnicity, and the tables in the supplement material are analyzed for genetically-defined ethnicity. Although the results are similar, is there a reason to set the analysis result for self-defined ethnicity as the main table?

2. East Asian (n=2464) was also defined in the supplement material, but why was it excluded from the analysis?

3. A total of 21 SNPs including APOE were selected, and the GRS excluding APOE also had an odds ratio >1. It seems that the GRS for other SNPs is also meaningful. Are there any related analysis results?

4. Typo(5-6 page) - The numbers in the quotation marks in the manuscript are different from Table 1- South Asian participants had the lowest frequency of ε2 alleles (“7.7%” with one and 0.2% with two alleles) and Black participants had the highest frequency (“14.8%” with one and 1.2% with two alleles) compared to the White participants (12.3% one and 0.6% two alleles). The ε4 allele was also most common in Black participants (“35.3%” one allele; 5.3% two alleles vs “17.3%” and 1.0% in South Asians and 26.3% and 2.4% in White participants).

5. “%” is not attached to (1.3) in the White British column of the Dementia cases N(%) row of Table

6. PLOS authors have the option to publish the peer review history of their article (what does this mean?). If published, this will include your full peer review and any attached files.

Reviewer #1: No

Reviewer #2: No

---

## [Author Response · Author response to Decision Letter 0]

27 Sep 2022

Reviewer #1: In the present manuscript, the authors used microarray data comprising 850,000 variants and more than 90 million variants were used for downstream analysis after imputation. After QC, they extracted previously identified dementia risk associated SNPs from previous GWAS (Lambert et al., 2013, Nature Genetics) which were not included in the UK Biobank set, and calculated GRS using those SNPs and weight in addition to APOE alleles. Since OR of APOE for dementia is extremely high, they used logOR for APOE alleles. Then they prepared 1) GRS with APOE alleles followed by z-standardization across whole sample (z-GRS with APOE), and 2) GRS without APOE alleles followed by z-standardization (z-GRS without APOE). They performed age, sex and ethnic group adjusted logistic regression analysis to investigate the effect of GRS and APOE alleles for dementia risk. Authors showed that significant increase dementia risk for APOE ε4 alleles, and the effects per SD increase of z-GRS with APOE and without APOE were significantly associated with increasing dementia risk, without significant ethnic group interaction effect. Authors also demonstrated association between SD increase of z-GRS and other features of dementia including family history, reaction time, and hippocampal and amygdala volumes, and showed significant association between z-GRS and family history. Also in this case, they have shown the effects of z-GRS for secondary outcomes had little or no significant interactions with other ethnic groups.

Overall, the manuscript falls below the level of publication in PLOS one, and authors are required to improve this study by referring the designs of many other research studies with strong understanding of genetics.

- Thank you for reviewing our manuscript. While some of the summary of our manuscript is correct, it is not correct that we used SNPs for the GRS that were not available within the UKBiobank samples. We used logOR as a weight for all SNPs and not just APOE in line with the accepted formula.

Major points

1)Main purpose of this study was to assess relevance of PRS model constructed from GWAS using Europeans. The way of showing applicability of European GWAS derived GRS in other ethnic groups is difficult to be agreed. Authors have measured interaction effect of ethnicity in logistic regression using z-GRS for dementia risk. This analysis presents whether ethnic differences interacted with z-GRS for dementia risk, not telling transferability of European GRS to South Asian and African groups. So this cannot support authors’ research questions. Instead, applying GRS model to South Asian and African separately, and then measure proportion of dementia cases or odds ratio by quantile groups, or area under receiver operating characteristics (AUROC) which represent classification power of the model.

- Testing interaction effects is a common and statistically appropriate way of measuring whether the impact of a given risk factor (in this case GRS) has a different impact on outcome (dementia) based on the value of a third variable (ethnicity in this case). This has been used often in interrogating whether the impact of risk factors varies by ethnic group (see for example, Eastwood SV, Tillin T, Chaturvedi N, Hughes AD. Ethnic Differences in Associations Between Blood Pressure and Stroke in South Asian and European Men : Novelty and Significance. Hypertension 2015; 66(3): 481-8). We have also now, as requested, calculated and plotted the AUROC curves for each ethnic group and provided these in the supplementary materials. The AUROC for the groups was 0.8236 for White, 0.8268 for South Asian and 0.8766 for Black participants. This shows the GRS using these SNPs performed very well in predicting dementia and was equally good in all ethnic groups, mirroring our interaction analyses.

2)Also authors conducted z-standardization to have z-score of GRS across whole sample in the UK Biobank set. This approach may diminish ethnic differences due to skewed proportion of White British (approximately 96% of total). It seems not appropriate to use z-standardization across whole sample for assessing transferability of European GRS to other ethnic groups.

- Our aim was to test whether GRS using GWAS derived from European samples would predict dementia in people from other ethnic groups. In order to explore the effect of ethnicity on the performance of the GRS, we needed to include all ethnic groups in the initial z-standardisation. If we only calculated z-score for each ethnic group separately this would have meant being unable to include ethnic group as a variable in our regression models and would have precluded us from examining ethnicity as an effect modifier.

3)To precisely assess applicability of European PRS models, various PRS calculation methods can be used. Authors can clump using linkage disequilibrium then adjusted p-value thresholds of European GWAS to construct various models (Pruning and thresholding) or leveraging genome-wide SNPs and adjust and shrink effect sizes using Bayesian models (PRScs or EB-PRS).

- Thank you for your comment. We agree that the methods could be used to generate PRS scores. However, we aimed to generate a GRS based solely on genome-wide significant hits from Lambert et al 2013. Therefore, we only included these 21 robustly associated SNPs. The SNPs were unlinked so did not require further processing. We opted for a limited GRS as we did not think that calculating a PRS would offer any advantage in this context. This approach has been used in several other studies, for example: doi.org/10.1212/WNL.0000000000200544

4)Authors are strongly required to graphically illustrate their result regarding their results.

- As requested we have now provided AUROC curves for all groups in the supplementary material

Minor points

1)Authors need to change the terms for APOE alleles; 0, 1 and 2 into non-carrier, heterozygous ε2 allele carrier, and homozygous ε2 allele carrier, respectively.

- We have now changes this in the text as suggested

2)In the Table 2, APOE allele non-carriers (0 in this paper) were labelled as “Reference” but this is not necessarily presented in the table, and exact p-values are missing for interaction effects.

- We have labelled non-carriers as reference category in the table and provided confidence intervals and p values in brackets in the table. We have copied the table below

 Main effect 

Exposure OR* LCI UCI p value Interaction effect, South Asian participants, OR (95%CI, p-value) Interaction effect, Black participants, OR (95%CI, p-value)

ε2 alleles 0 Reference OR 1.46 (0.58-3.66, 0.336) OR 0.95 (0.47-1.95, 0.896)

 1 0.54 0.48 0.60 <0.0001 

 2 0.56 0.35 0.89 0.014 

ε4 alleles 0 Reference OR 0.92 (0.60-1.39, 0.676) OR 0.99 (0.72-1.37, 0.974)

 1 2.59 2.44 2.75 <0.0001 

 2 8.47 7.68 9.34 <0.0001 

z-GRS with APOE - effect per SD increase 1.73 1.69 1.77 <0.0001 OR 0.97 (0.76-1.20, 0.687) OR 0.99 (0.83-1.17, 0.864)

z-GRS without APOE – effect per SD increase 1.21 1.18 1.24 <0.0001 OR 0.82 (0.65-1.05, 0.113) OR 0.94 (0.72-1.23, 0.620)

3)Authors have used threshold for Info score of 0.9, and this might be too high. Authors may present references for justifying this threshold.

- The Info score is a measure of imputation reliability and we wished to only include SNPs which we could be relatively certain were imputed accurately. We therefore chose the threshold of 0.9 to avoid including poor quality data. This threshold is at the discretion of researchers but we preferred to err on the side of caution. See for example https://doi.org/10.1371/journal.pone.0009697

4)Authors are required to proofread the manuscript to avoid awkward descriptions.

- We have now done this but are happy to edit specific sections if you felt these were unclear.

5)Result section should be divided into subtitles for better readability.

 

- We have now added subheadings as suggested.

Reviewer #2: Through Genome-Wide Association Studies (GWAS), researchers have developed Genetic Risk Scores (GRS) for numerous diseases. However, these studies are mostly biased towards European white races, so their effectiveness in other populations is not certain. Recognizing this problem, the authors verified the GRS developed for dementia in other races other than European white.

Such verification is expected to improve reliability in the use of the developed GRS as well as to enable the classification of GRS that cannot be used for various races. Therefore, I agree to accept this research for publication on Plos One.

- Thank you for reviewing our manuscript.

I found a few minor things that needed to be corrected as described below.

1. The tables in the paper are analyzed for self-defined ethnicity, and the tables in the supplement material are analyzed for genetically-defined ethnicity. Although the results are similar, is there a reason to set the analysis result for self-defined ethnicity as the main table?

- We chose self-defined ethnicity for our primary analysis as this is more easily available for most studies and is a practical choice for grouping. We have stated this in our methods section.

2. East Asian (n=2464) was also defined in the supplement material, but why was it excluded from the analysis?

- The QC procedures for non-European participants identified an East Asian group and is shown in the supplementary material for completeness. However, we pre-specified we would use self-defined ethnic group and compare between the three largest ethnic groups in the sample which were White, South Asian and Black only.

3. A total of 21 SNPs including APOE were selected, and the GRS excluding APOE also had an odds ratio >1. It seems that the GRS for other SNPs is also meaningful. Are there any related analysis results?

- We are not sure what related analyses you mean. We have included all results including regression analyses for primary and secondary outcomes with and without APOE and using self-defined ethnicity and genetic ancestry grouping.

4. Typo(5-6 page) - The numbers in the quotation marks in the manuscript are different from Table 1- South Asian participants had the lowest frequency of ε2 alleles (“7.7%” with one and 0.2% with two alleles) and Black participants had the highest frequency (“14.8%” with one and 1.2% with two alleles) compared to the White participants (12.3% one and 0.6% two alleles). The ε4 allele was also most common in Black participants (“35.3%” one allele; 5.3% two alleles vs “17.3%” and 1.0% in South Asians and 26.3% and 2.4% in White participants).

- thank you for noticing this. We have now corrected the numbers in the text.

5. “%” is not attached to (1.3) in the White British column of the Dementia cases N(%) row of Table

- We have now removed the % sign from the other numbers as we have indicated in the first column that all numbers in brackets are percentages

---

## [Decision Letter · Decision Letter 1]

26 Oct 2022

Genetic risk scores and dementia risk across different ethnic groups in UK Biobank

PONE-D-22-19235R1

Dear Dr. Naaheed Mukadam,

We’re pleased to inform you that your manuscript has been judged scientifically suitable for publication and will be formally accepted for publication once it meets all outstanding technical requirements.

Kind regards,

Jeong-Sun Seo

Academic Editor

PLOS ONE

Additional Editor Comments (optional):

Reviewers' comments:

Reviewer's Responses to Questions

**Comments to the Author**

1. If the authors have adequately addressed your comments raised in a previous round of review and you feel that this manuscript is now acceptable for publication, you may indicate that here to bypass the “Comments to the Author” section, enter your conflict of interest statement in the “Confidential to Editor” section, and submit your "Accept" recommendation.

Reviewer #2: All comments have been addressed

2. Is the manuscript technically sound, and do the data support the conclusions?

Reviewer #2: Yes

3. Has the statistical analysis been performed appropriately and rigorously? 

Reviewer #2: Yes

4. Have the authors made all data underlying the findings in their manuscript fully available?

Reviewer #2: Yes

5. Is the manuscript presented in an intelligible fashion and written in standard English?

Reviewer #2: Yes

6. Review Comments to the Author

Reviewer #2: The authors have addressed all my comments satisfactorily. I would recommend this manuscript to move towards publication.

7. PLOS authors have the option to publish the peer review history of their article (what does this mean?). If published, this will include your full peer review and any attached files.

Reviewer #2: No

---

## [Editor Report · Acceptance letter]

11 Nov 2022

PONE-D-22-19235R1 

Genetic risk scores and dementia risk across different ethnic groups in UK Biobank 

Dear Dr. Mukadam:

I'm pleased to inform you that your manuscript has been deemed suitable for publication in PLOS ONE. Congratulations! Your manuscript is now with our production department. 

Kind regards, 

on behalf of

Dr. Jeong-Sun Seo 

Academic Editor

PLOS ONE